# Determination of Virulence-Associated Genes and Antimicrobial Resistance Profiles in *Brucella* Isolates Recovered from Humans and Animals in Iran Using NGS Technology

**DOI:** 10.3390/pathogens12010082

**Published:** 2023-01-03

**Authors:** Maryam Dadar, Saeed Alamian, Hanka Brangsch, Mohamed Elbadawy, Ahmed R. Elkharsawi, Heinrich Neubauer, Gamal Wareth

**Affiliations:** 1Razi Vaccine and Serum Research Institute (RVSRI), Agricultural Research, Education and Extension Organization (AREEO), Karaj 3197619751, Iran; 2Friedrich-Loeffler-Institute, Institute of Bacterial Infections and Zoonoses (IBIZ), 96a, D-07743 Jena, Germany; 3Laboratory of Veterinary Pharmacology, Faculty of Agriculture, Tokyo University of Agriculture and Technology, 3-5-8 Saiwai-cho, Fuchu, Tokyo 183-8509, Japan; 4Department of Pharmacology, Faculty of Veterinary Medicine, Benha University, Toukh 13736, Egypt; 5Internal Medicine III, Tropical Medicine, Tanta University Hospital, Tanta 31527, Egypt

**Keywords:** *B. melitensis*, *B. abortus*, antimicrobial-resistant, virulence genes, WGS

## Abstract

Brucellosis is a common zoonotic disease in Iran. Antimicrobial-resistant (AMR) *Brucella* isolates have been reported from different developing countries, posing an imminent health hazard. The objective of this study was to evaluate AMR and virulence-associated factors in *Brucella* isolates recovered from humans and animals in different regions of Iran using classical phenotyping and next generation sequencing (NGS) technology. Our findings revealed that *B. melitensis* is the most common species in bovines, small ruminants and camels. *B. abortus* was isolated only from one human case. Probable intermediate or resistant phenotype patterns for rifampicin, trimethoprim-sulfamethoxazole, ampicillin-sulbactam and colistin were found. Whole genome sequencing (WGS) identified *mprF*, *bepG*, *bepF*, *bepC*, *bepE*, and *bepD* in all isolates but failed to determine other classical AMR genes. Forty-three genes associated with five virulence factors were identified in the genomes of all *Brucella* isolates, and no difference in the distribution of virulence-associated genes was found. Of them, 27 genes were associated with lipopolysaccharide (LPS), 12 genes were related to a type IV secretion system (*virB1-B12*), two were associated with the toll-interleukin-1 receptor (TIR) domain-containing proteins (*btpA*, *btpB*), one gene encoded the Rab2 interacting conserved protein A (*ricA*) and one was associated with the production of cyclic β-1,2 glucans (*cgs*). This is the first investigation reporting the molecular-based AMR and virulence factors in brucellae isolated from different animal hosts and humans in Iran. Iranian *B. abortus* and *B. melitensis* isolates are still in vitro susceptible to the majority of antibiotics used for the treatment of human brucellosis. WGS failed to determine classical AMR genes and no difference was found in the distribution of virulence-associated genes in all isolates. Still, the absence of classical AMR genes in genomes of resistant strains is puzzling, and investigation of phenotypic resistance mechanisms at the proteomic and transcriptomic levels is needed.

## 1. Introduction

Brucellosis is a zoonotic bacterial infection causing significant economic losses in livestock populations and severely debilitating disease in humans worldwide [1,2]. Brucellosis can be transmitted from infected animals to humans mainly by consuming unpasteurized dairy and undercooked meat/meat products, or through close contact with an infected animal [3]. Four species of the genus *Brucella* (*B*.) are pathogenic for humans: *B. melitensis* (main hosts: sheep and goat), *B. suis* (swidae), *B. abortus* (bovidae), and *B. canis* (canidae) [4]. However, the cross-species infection of hosts with different *Brucella* spp. has also been documented [5]. The clinical symptoms of human brucellosis are non-specific and variable, ranging from severe acute or irregular febrile illness, headache, anorexia, fatigue, weight loss, generalized aching and arthralgia to chronic disease with severe complications [6]. The treatment of brucellosis demands prolonged and appropriate antimicrobial therapy. The antibiotic treatment of intracellular bacteria such as *Brucella* is hampered by handicapped intracellular diffusion of antibiotics as well as the development of resistance to antibiotics [7,8,9]. Different mechanisms can induce antibiotic resistance, e.g., efflux pumps, enzymatic inactivation, horizontal gene transfer and modification of drug targets [10]. In several studies, antibiotic resistance to commonly used antimicrobial drugs, including trimethoprim/sulfamethoxazole and rifampicin, has been reported [8,11,12]. Therefore, susceptibility testing and detection of genetic resistance determinants have been strongly proposed as effective methods to monitor the efficiency of antibiotics for the advised treatment of human brucellosis [13,14,15]. However, antimicrobial susceptibility testing is not carried out routinely due to the traditional approach to therapy and technical challenges. It has to be noted that laboratory infections with aerosolized *Brucella* spp. often occur; thus, a specialized BSL3 laboratory needs to be established [16]. In Iran, several studies on brucellosis showed that *B. abortus* and *B. melitensis* were the most isolated *Brucella* species in livestock and humans [17,18]. Most human brucellosis cases have been caused by *B. melitensis*, and a few by *B. abortus* [19]. Different studies have reported a relapse rate of 13–18% in Iranian patients as one of the most critical complications, even following an appropriate treatment [20,21,22]. Several studies reported AMR in a few isolates [12,23] and an increasing trend in the number of *Brucella* isolates with a resistant phenotype was noted [24,25]. However, it is unclear whether this is due to the development of intrinsic or acquired resistance against antibiotic compounds, or whether the results are a consequence of the intracellular nature of brucellae and thus the inability of antimicrobial compounds to penetrate the infected site, e.g., bone tissue or reticuloendothelial cells. Furthermore, identification and characterization of virulent genes are essential to implementing efficient disease control and prevention approaches, and evaluating the pathogenicity of brucellae [26]. However, very few investigations have been performed applying whole-genome sequencing (WGS) to evaluate the antimicrobial resistance and investigate virulence genes in *Brucella* worldwide [27,28]. No data on the resistance genes of *Brucella* isolates based on WGS exist from Iran. The current study aimed to examine the sensitivity of *Brucella* isolates recovered from humans and animals in Iran against most of the antibiotics used for brucellosis treatment in human patients to verify the adequacy of current treatment guidelines. Moreover, NGS technology was applied to investigate AMR and virulence-associated genes.

## 2. Materials and Methods

### 2.1. Ethics Committee

This survey was part of the national surveillance plan for brucellosis, 2015–2020, and all the activities follow the ethics requirement of the plan. This study was approved by the ethics committee of the Razi Vaccine and Serum Research Institute, Karaj, Iran (IR.RVSRI.REC.2015.001) in 2015, confirming that all experiments were performed following relevant guidelines and regulations. The patients gave informed consent for sampling and for participating in the survey/questionnaire.

### 2.2. Brucella Isolates

Forty *Brucella* isolates (23 of human origin and 17 of animal origin) from culture-positive human and animal cases of brucellosis collected at the Razi Vaccine and Serum Research Institute, Karaj, Iran from 2015 to 2020 were analyzed. For this study, we selected the exanimated isolates according to the various species, various biovars and different geographical locations that *Brucella* spp. isolated in Iran. The isolates were recovered from different specimens. Human isolates (*n* = 23) were recovered from blood samples (*n* = 22) and one cerebrospinal fluid (CSF) sample. Animal isolates (*n* = 17) were recovered from milk (six cows, one camel, and one sheep), four lymph nodes (three cows and one camel), and five aborted fetuses (four sheep and one goat). All camels were apparently healthy and had not received any *Brucella* vaccine. Goats, cows, and sheep had an abortion history on the farm. Animal samples from aborted fetuses (abomasum content, liver, spleen, and kidneys) and milk were gathered in a sterile falcon tube and kept at −20 °C until further evaluation. Human cases were patients referred with clinical complaints of brucellosis to Razi laboratory with positive Rose Bengal, Wright, and 2ME tests. 

### 2.3. Brucella Isolation, Biotyping, and Molecular Confirmation

For bacterial isolation, milk samples were centrifuged for 15 min at 3000× *g*. Subsequently, the sediment and the creamy upper layer of samples were spread on cultivation media. Primary cultivation of all samples was carried out by streaking of 10 μL samples on a *Brucella* selective agar [*Brucella* agar (Himedia, Mumbai, India) supplemented with 5% inactivated horse serum, nystatin (50,000 IU), nalidixic acid (2.5 mg), vancomycin (10 mg), bacitracin (12,500 IU), polymyxin B (2500 IU), and cycloheximide (50 mg) (Oxoid, Basingstoke, UK)]. Plates were kept for 14 days at 37 °C under 10% CO_2_. Suspected colonies, i.e., round, pinpoint, translucent, and pearly white colonies were selected for further analysis. A panel of classical biotyping tests was performed, i.e., H2S production, dependence on carbon dioxide (CO_2_), lysis by specific phages, agglutination by acriflavine, growth characteristics on dye-agar media containing thionin and basic fuchsin and agglutination with specific monospecific *Brucella* antisera of A and M [29]. The interpretation of results was performed according to the WOAH (OIE) manual (http://www.oie.int/en/animal-health-in-the-world/animal-diseases/Brucellosis/ (accessed on 15 October 2022)). Extraction of genomic DNA was carried out using the Exgene Cell SV kit (GeneAll, Seoul, Republic of Korea) according to the manufacturers protocol. Nanodrop (Thermo Scientific, Waltham, MA, USA) was used to evaluate DNA concentration. Furthermore, the DNA integrity was analyzed with 1% agarose gel. Samples were stored at −20 °C for later analysis. AMOS-PCR and Bruce-Ladder PCR were done as previously described [30,31]. 1% agarose gel electrophoresis was used to resolve the PCR products.

### 2.4. Antibiotic Susceptibility Testing (AST)

All identified isolates were subjected to AST using disk diffusion susceptibility tests and minimum inhibitory concentrations (MICs) tests for nine antibiotics. Disk diffusion susceptibility tests were performed for the antibiotics gentamicin (10 μg), streptomycin (10 μg), rifampin (5μg), doxycycline (30 μg), ceftriaxone (30 μg), ampicillin-sulbactam (10 + 10 μg) and trimethoprim/sulfamethoxazole (1.25/23.75 μg). The criteria for choosing these nine antibiotics was based on WHO guidelines on treatment of human brucellosis and the recommendation of used antibiotics in different studies [25]. The results of all antimicrobial tests were read after 48 h. The thresholds of antibiotic tests were set using the guidelines for *Haemophilus* spp. [8,32,33]. The MICs were evaluated by E-test. Briefly, a solution (0.5 McFarland standard) was prepared and spread onto the Muller-Hinton agar plates enriched with 5% sheep’s blood. The bacterial plates were kept under 10% CO_2_ at 37 °C, and AST was recorded after 48 h. MICs of bacterial isolates for rifampin (0.016–256 μg/mL), gentamicin (0.064–1024 μg/mL), doxycycline (0.016–256 μg/mL), ceftriaxone (0.016–256 μg/mL), streptomycin (0.064–1024 μg/mL), trimethoprim/sulfamethoxazole (0.002–32 μg/mL), imipenem (0.002–32 μg/mL), ampicillin (0.016–256 μg/mL), and colistin (0.016–256 μg/mL) were evaluated according to the manufacturer’s protocol (liofilchem/Italy) and the Clinical and Laboratory Standards Institute (CLSI) guidelines (2020) [34]. Reference strains, *Pseudomonas aeruginosa* (ATCC 27853), *Escherichia coli* (ATCC 25922), *Staphylococcus aureus* (ATCC 29213), *Streptococcus pneumonia* (ATCC 49619), and *Enterococcus faecalis* (ATCC 29212) were used to confirm the AST results. All ASTs were performed in duplicates for all *Brucella* isolates [35,36].

### 2.5. WGS and in Silico Detection of AMR and Virulence-Associated Genes

After DNA extraction through the Exgene Cell SV kit (GeneAll, South Korea), the sequencing library was prepared. The samples were sequenced on an Illumina MiSeq (Illumina, San Diego, CA, USA) using paired-end sequencing. The analysis and assembly of raw sequencing data were carried out as previously described [27]. Detection of the gene and protein sequences for AMR-associated genes was carried out through several databases, including the Resistance Gene Identifier (RGI) according to the ResFinder database [37], the Comprehensive Antibiotic Resistance Database (CARD) [38], and the NCBI AMR Finder Plus (https://github.com/ncbi/amr/wiki/Running-AMRFinderPlus, accessed on 15 October 2022) [39]. Potential virulence-associated genes were identified via the virulence factor database using the core dataset “VFDB, http://www.mgc.ac.cn/VFs/ (accessed on 22 June 2022)” [27,40].

## 3. Results

### 3.1. Brucella Identification and Characterization

In the current study, 40 *Brucella* isolates (3 *B. abortus* and 37 *B. melitensis*) strains were characterized by cultivation-dependent and molecular methods (Bruce-Ladder PCR, AMOS PCR, and WGS). The results of Bruce-Ladder PCR and AMOS PCR were consistent with the results of WGS. *B. melitensis* was detected in human blood (*n* = 21), and human CSF (*n* = 1), bovine milk (*n* = 4), bovine lymph nodes (*n* = 3), camel milk (*n* = 1), camel lymph nodes (*n* = 1), and ovine aborted fetus (*n* = 4), ovine milk (*n* = 1) and caprine aborted fetus (*n* = 1) samples. These isolates showed identical results in the Bruce-Ladder PCR and AMOS PCR, and were also identified as *B. melitensis* through WGS. One isolate from a human patient and two isolates from milk of seropositive cows were confirmed as *B. abortus* by PCR and WGS using the Kraken program (Table 1).

### 3.2. Phenotypic AMR Profiles of Brucella Strains

The obtained MIC and disk diffusion values for all tested antibiotics are shown in Table 2 and Table 3. According to MIC measurements, all tested *Brucella* isolates were susceptible to doxycycline (MIC90 = 0.094 μg/mL), rifampicin (MIC90 = 0.5 μg/mL), gentamycin (MIC90 = 0.75 μg/mL), imipenem (MIC90 = 4 μg/mL), ceftriaxone (MIC90 = 0.75 μg/mL), streptomycin (MIC90 = 0.5 μg/mL) and trimethoprim-sulfamethoxazole (MIC90 = 0.064 μg/mL). Non-susceptible isolates (resistant and intermediate) were observed only for colistin and ampicillin-sulbactam. All *B. abortus* and *B. melitensis* isolates (*n* = 40) showed resistance to colistin, while most *B. melitensis* isolates (*n* = 23) had intermediate resistance for ampicillin-sulbactam (MIC90 = 2 μg/mL) (Table 2). In disk diffusion assays, 12 *B. melitensis* isolates (32.5%) showed an inhibition zone of 19–17 mm for discs with 5 μg rifampicin. They were classified as intermediate rifampin-resistant according to the slow-growing bacteria standards of CLSI. In contrast, all *B. abortus* isolates (*n* = 3) and 17 *B. melitensis* isolates exhibited an inhibition zone of ≥20 mm (20 and 33 mm, respectively) and were considered susceptible. Resistance to rifampicin was seen in 8 *B. melitensis* (21.6%) isolates by disk diffusion values of ≤16 μg/mL for incubation at 10% CO_2_. In contrast, only 13 *B. melitensis* (35%) isolates showed resistance to ampicillin-sulbactam according to disk diffusion values and breakpoints of between 13 and 19 mm (≤19 mm) (Table 3).

### 3.3. Whole-Genome Sequencing and Data Availability

Sequencing of 40 Iranian *Brucella* strains yielded an average of 1,645,251 reads per isolate (range 1,217,718–2,835,032) with an average length of 275 bp. The mean coverage of genome sequences was 105.9, ranging from 99 to 202. Moreover, the Kraken2 software was applied to classify each read and assembled contig to evaluate the accurate species identification and detection of potential contaminations [41]. The first match for all isolates at the genus level was “*Brucella*”, on average 99.7% of the reads (minimum 99.5%, maximum 99.8%). At the level of species detection, the first match for 37 isolates was “*B. melitensis*”, and three isolates were confirmed as “*B. abortus*”. From these reads, genomes were assembled for all isolates with an average genome size of 3288,126 bp, minimum of 3243,150 bp and a mean N50 of 336.295 bp (range 251,030–462,204 bp). The GC ratio was on average 57.24% (Appendix A). All study data are included in the article and supporting information (Appendix A). The data have also been submitted to the European Nucleotide Archive (ENA). The project accession number is PRJEB50179.

### 3.4. In Silico Identification of AMR and Virulence-Associated Genes

The in silico analysis of AMR genes in 40 genomes of Iranian *Brucella* isolates using several databases yielded only the multiple peptide resistance factors (*Brucella_suis*_mprF) protein and efflux-related genes *bepC*, *bepD*, *bepE*, *bepF*, *bepG* by the CARD and AMRFinderPlus, respectively (Appendix A). Those genes were found in all *Brucella* genomes, either susceptible or resistant, except one strain lacked *bepG* and *bepF*. The virulence factor database (VFDB) revealed the presence of forty-three virulence and pathogenicity factors in all tested *Brucella* strains (Table 4). The majority of these were associated with the LPS (lipopolysaccharide) operon (*n* = 27), followed by genes encoding the type IV secretion system (*virB1-B12*). Furthermore, two genes (*btpA*, *btpB*) code for TIR domain-containing proteins that inhibit dendritic cell maturation and proinflammatory cytokines’ production, increasing immune evasion. One gene, *ricA*, encoded the Rab2 interacting conserved protein A that specifically interacts with the GDP-bound form of Rab2 and may play an influential role in the maturation of the *Brucella*-containing vacuole, as it could slow down intracellular replication and thus increase evasion from the innate immune system. Finally, one gene (*cgs*) is associated with the production of cyclic β-1,2-glucans that increase intracellular survival (Table 4). It is significant to highlight that Iranian *B. abortus* and *B. melitensis* isolates showed no difference in the distribution of virulence-associated genes, even from different hosts.

## 4. Discussion

Brucellosis remains a notorious zoonotic infection causing significant damage to the farming industry and public health. The disease is prevalent in Mediterranean and Middle Eastern countries including Iran [42,43,44]. According to the WHO, the brucellosis burden specifically on developing and low-income countries substantiates its classification as a serious zoonotic disease. In Iran, *B. melitensis* and *B. abortus* are the dominant *Brucella* species [19]. Our study confirmed the accurate diagnosis of *Brucella* spp. isolated from animals and humans through classical and molecular typing. The current study used classical biotyping and bacterial culture as the gold standard combined with DNA-based tools (multiplex PCR and WGS) to identify *Brucella* strains isolated from various hosts. The diagnosis of the different methods used in this study was consistent at the genus and species levels. Multiplex PCR tests and WGS appeared to be a reliable and rapid approach for the accurate classification of *Brucella* strains [27,31] with the potential to replace classical biotyping. *Brucella* identification by molecular methods has been reported as a fast and precise test in diagnostic laboratories. However, for improving in silico *Brucella* identification, global reference databases are required to identify different species accurately. In this way, WGS provides a powerful method for accurate typing of *Brucella* spp. because of the evaluation of the entire bacterial genome, thus improving discriminatory power [45,46] and replacing or overcoming the classical approach. Although WGS enables detailed strain typing, PCR should still be considered an essential method. It quickly provides information on genus and species identity and thus helps to take appropriate safety measures to decrease the risk of laboratory-acquired infections. These results improve our knowledge on the current species of *Brucella* in ruminant and human reservoirs of Iran and confirm a significant burden of *B. melitensis*. In livestock, *B. melitensis* is common in bovines, camels, and small ruminants [44]. The increasing number of cases of *B. melitensis* in cattle are also reported from other Middle East and African countries [47,48,49,50] and Iran [51]. The results of this study confirm that human brucellosis in Iran is mainly caused by *B. melitensis*, which is the predominant species causing human disease globally [52,53,54].

In Iran, brucellosis treatment follows the World Health Organization (WHO) guidelines [6]. The WHO recommends a combination of streptomycin and doxycycline or gentamicin and doxycycline for patients younger than 60 years, and rifampicin and doxycycline for patients older than 60 years and children [55,56]. For pregnant women, trimethoprim/sulfamethoxazole (cotrimoxazole) plus rifampin has been used [57]. For pregnant women <36 weeks gestation, the treatment depends on a combination of trimethoprim/sulfamethoxazole and rifampicin, and after 36 weeks gestation, rifampicin is administered as a monotherapy [58]. Our results show that all *Brucella* isolates were susceptible to ceftriaxone, imipenem, doxycycline and gentamicin. The findings that *Brucella* isolates were susceptible to tetracycline, doxycycline, streptomycin, ciprofloxacin, gentamicin, levofloxacin and trimethoprim-sulfamethoxazole are in accordance with reports from Egypt [27], Turkey [59], Saudi Arabia [60], China [61] and Norway [28]. Intermediate or resistant phenotypes for rifampicin, trimethoprim-sulfamethoxazole, ampicillin-sulbactam and colistin were, however, found. Indeed, antimicrobial resistance is frequently observed in brucellae [62]. This increase in AMR may be responsible for the increasing number of relapses which was reported over the last few years in different studies [20,35,61]. However, AST is often not practised before the start of the treatment due to the notorious serological diagnosis and the lack of suitable samples. A considerable number of brucellae were found resistant to rifampin in Iran [23,63] and several countries in the Middle East, such as Turkey [9,15], Saudi Arabia [64], Qatar [65] and Egypt [27]. However, it is still an essential antibiotic in the treatment regimens of brucellosis. Trimethoprim-sulfamethoxazole was found to be an effective antimicrobial compound in the treatment of human brucellosis [66]. However, a decreased susceptibility was reported in Iran previously [32,63,67,68].

The analysis of WGS data of 40 Iranian isolates revealed two AMR genes, multiple peptide resistance factors (*mpr*F) and the outer membrane efflux protein *bep* G, F, C, E and D in all strains. It is known that *mpr*F plays an essential role in resistance to cationic antibiotics such as gentamycin, moenomycin and vancomycin [69]. However, the results from the disk diffusion assays in this study showed no resistance to gentamycin. It is known that *bep* proteins increase resistance to some antibiotic compounds such as tetracycline, doxycycline, chloramphenicol and ciprofloxacin in *B. suis* [70]. Hence, none of these *B. abortus* and *B. melitensis* isolates showed resistance to those antibiotics. The inability to detect classical resistance genes in the genome of brucellae is puzzling. This finding might be explained by the presence of other factors, like mutations in housekeeping genes or regulatory mechanisms [71], or till now unknown AMR genes in *Brucella*, which are not yet registered in public AMR databases. Furthermore, the intracellular lifestyle of brucellae that prohibits the penetration of different antimicrobials into the cells may play a role in the resistance development of these bacteria. There are only a few molecular-based investigations on the genetic determinants of antibiotic resistance in brucellae [28,61,72,73].

In the present study, we have also analyzed the virulence genes of different *Brucella* isolates in silico. Brucellae, as facultative intracellular bacteria, do not use “classic” virulence factors such as proteases, cytolysins, exotoxins, capsules, exoenzymes, virulence plasmids and pili or fimbriae [74]. In the current study, most identified pathogenicity-associated genes are involved in LPS production and type IV secretion systems. Until now, there exist few reports on the detection of virulence-associated genes in *Brucella* strains isolated from humans and livestock from Iran [75,76]. Examination of *B. abortus* and *B. melitensis* strains isolated from animal and human hosts in Iran revealed the presence of *virB5*, *btpA*, *btpB*, *vceC*, *bpe275*, *bspB*, and *virB2* genes in all strains, while *betB* was found in 97% and *prpA* in 86% of the strains [75]. Another study also reported the presence of *ure*, *wbkA*, *omp19*, *manA, mviN*, and *perA* genes in *B. melitensis* and *B. abortus* using multiplex-PCR tests [77]. Although several PCR methods have been reported to identify virulence- and resistance-associated genes in *Brucella* isolates, these methods are limited by the species-specificity of used primers.

## 5. Conclusions

The implementation of high-throughput WGS allowed for more comprehensive detection of virulence- and resistance-associated genes. No clear difference in the distribution of the AMR and virulence genes among both resistant and sensitive *B. abortus* and *B. melitensis* strains was found, even for those recovered from different hosts. Therefore, further investigations of antibiotic susceptibility have to be continued on *Brucella* isolates. Although the study of resistance and virulence mechanisms based on the genome was helpful and provided a comprehensive explanation in several microorganisms, it is of little value in the case of brucellae. Thus, resistance and virulence mechanisms at the proteomic and transcriptomic levels have to be considered in brucellae in future research.

## Figures and Tables

**Table 1 pathogens-12-00082-t001:** Molecular characterization of *B. melitensis* and *B. abortus* isolates from humans and ruminants in Iran.

ID	Host	Source	Year	Location	Description	Biotyping	PCR	WGS
RAZI20Y0140	human	blood	2015	Alborz	♂, farmer	*B. abortus*	*B. abortus*	*B. abortus*
RAZI20Y0141	human	blood	2015	Tehran	♂, farmer	*B. melitensis*	*B. melitensis*	*B. melitensis*
RAZI20Y0142	human	blood	2016	Alborz	♀, farmer	*B. melitensis*	*B. melitensis*	*B. melitensis*
RAZI20Y0143	human	blood	2017	Tehran	♂, farmer	*B. melitensis*	*B. melitensis*	*B. melitensis*
RAZI20Y0144	human	blood	2018	Kermanshah	♀, farmer	*B. melitensis*	*B. melitensis*	*B. melitensis*
RAZI20Y0145	human	CNS	2015	Alborz	♀, retired	*B. melitensis*	*B. melitensis*	*B. melitensis*
RAZI20Y0146	human	blood	2020	Tehran	♂, farmer	*B. melitensis*	*B. melitensis*	*B. melitensis*
RAZI20Y0147	human	blood	2015	Kerman	♂, farmer	*B. melitensis*	*B. melitensis*	*B. melitensis*
RAZI20Y0148	human	blood	2020	Alborz	♂, teacher	*B. melitensis*	*B. melitensis*	*B. melitensis*
RAZI20Y0149	cow	milk	2015	Qom	abortion	*B. melitensis*	*B. melitensis*	*B. melitensis*
RAZI20Y0150	cow	milk	2017	Tehran	abortion	*B. melitensis*	*B. melitensis*	*B. melitensis*
RAZI20Y0151	cow	milk	2016	Qom	abortion	*B. melitensis*	*B. melitensis*	*B. melitensis*
RAZI20Y0152	camel	milk	2017	Tehran	seropositive	*B. melitensis*	*B. melitensis*	*B. melitensis*
RAZI20Y0153	cow	milk	2018	Yazd	abortion	*B. abortus*	*B. abortus*	*B. abortus*
RAZI20Y0154	cow	milk	2019	Fars	abortion	*B. melitensis*	*B. melitensis*	*B. melitensis*
RAZI20Y0155	sheep	milk	2018	Mazandaran	abortion	*B. melitensis*	*B. melitensis*	*B. melitensis*
RAZI20Y0156	cow	milk	2019	Fars	abortion	*B. abortus*	*B. abortus*	*B. abortus*
RAZI20Y0157	human	blood	2018	Kermanshah	♂, farmer	*B. melitensis*	*B. melitensis*	*B. melitensis*
RAZI20Y0158	human	blood	2018	Kermanshah	♀, house- keeper	*B. melitensis*	*B. melitensis*	*B. melitensis*
RAZI20Y0159	human	blood	2019	Alborz	♂, farmer	*B. melitensis*	*B. melitensis*	*B. melitensis*
RAZI20Y0160	human	blood	2019	Kermanshah	♂, farmer	*B. melitensis*	*B. melitensis*	*B. melitensis*
RAZI20Y0161	human	blood	2019	Alborz	♀, farmer	*B. melitensis*	*B. melitensis*	*B. melitensis*
RAZI20Y0162	human	blood	2019	Tehran	♂, farmer	*B. melitensis*	*B. melitensis*	*B. melitensis*
RAZI20Y0163	human	blood	2019	Hamedan	♂, farmer	*B. melitensis*	*B. melitensis*	*B. melitensis*
RAZI20Y0164	human	blood	2019	Hamedan	♀, farmer	*B. melitensis*	*B. melitensis*	*B. melitensis*
RAZI20Y0165	human	blood	2019	Hamedan	♀, house- keeper	*B. melitensis*	*B. melitensis*	*B. melitensis*
RAZI20Y0166	human	blood	2019	Kermanshah	♂, farmer	*B. melitensis*	*B. melitensis*	*B. melitensis*
RAZI20Y0167	human	blood	2019	Alborz	♂, farmer	*B. melitensis*	*B. melitensis*	*B. melitensis*
RAZI20Y0168	human	blood	2019	Alborz	♀, farmer	*B. melitensis*	*B. melitensis*	*B. melitensis*
RAZI20Y0169	human	blood	2019	Kermanshah	♂, farmer	*B. melitensis*	*B. melitensis*	*B. melitensis*
RAZI20Y0170	human	blood	2019	Tehran	♀, farmer	*B. melitensis*	*B. melitensis*	*B. melitensis*
RAZI20Y0171	sheep	aborted fetus	2020	Fars	abortion	*B. melitensis*	*B. melitensis*	*B. melitensis*
RAZI20Y0172	sheep	aborted fetus	2020	Yazd	abortion	*B. melitensis*	*B. melitensis*	*B. melitensis*
RAZI20Y0173	cow	L.N	2020	Fars	abortion	*B. melitensis*	*B. melitensis*	*B. melitensis*
RAZI20Y0174	cow	L.N	2019	Semnan	abortion	*B. melitensis*	*B. melitensis*	*B. melitensis*
RAZI20Y0175	cow	L.N	2019	Isfahan	abortion	*B. melitensis*	*B. melitensis*	*B. melitensis*
RAZI20Y0176	sheep	aborted fetus	2019	Zanjan	abortion	*B. melitensis*	*B. melitensis*	*B. melitensis*
RAZI20Y0177	goat	aborted fetus	2019	Alborz	abortion	*B. melitensis*	*B. melitensis*	*B. melitensis*
RAZI20Y0178	sheep	aborted fetus	2020	Fars	abortion	*B. melitensis*	*B. melitensis*	*B. melitensis*
RAZI20Y0179	camel	L.N	2020	Hormozgan	seropositive	*B. melitensis*	*B. melitensis*	*B. melitensis*

Male ♂; Female ♀.

**Table 2 pathogens-12-00082-t002:** MIC values of antibiotics against human and ruminant *Brucella* isolates using E-test.

Antibiotics	MIC Range (μg/mL)	MIC Range 50 (μg/mL)	MIC Range 90 (μg/mL)	MIC Interpretive Criteria (μg/mL)		
				S	R	I
Ceftriaxone	0.032–1	0.25	0.75	≤2	-	-
Imipenem	1.5–8	2	4	≤4	-	-
Doxycycline	0.032–0.125	0.064	0.094	≤4	8	≥16
Rifampicin	0.047–0.75	0.38	0.5	≤1	2	≥4
Streptomycin	0.094–0.75	0.38	0.5	≤8	-	-
Colistin	R	R	R	ND	-	-
Trimethoprim-Sulfamethoxazole	0.023–0.064	0.047	0.064	≤0.5	1–2	≥4
Gentamycin	0.094–1	0.38	0.75	≤4	-	-
Ampicillin-sulbactam	0.25–3	1.5	2	≤1	2	≥4

Standard breakpoints according to the guidelines for slowly growing bacteria (*Haemophilus* spp.) from CLSI S: Sensitive; I: Intermediate, and R: Resistant. ND: not described by CLSI standards.

**Table 3 pathogens-12-00082-t003:** Antibiotic susceptibility testing of human and ruminant *Brucella* isolates using disk diffusion testing.

Antibiotics	Concentration μg/disk	Range (mm)	Sensitive no (%)	Intermediate no (%)	Resistant no (%)	Resistance Pattern		
						S	I	R
Ceftriaxone	30 μg	25–62	40 (100)	0	0	≥26	ND	ND
Imipenem	10 μg	21–39	40 (100)	0	0	≥16	ND	ND
Doxycycline	30 μg	29–48	40 (100)	0	0	10≥	ND	ND
Rifampicin	5 μg	15–33	20 (50)	12 (30%)	8 (20%)	≥20	17–19	≤16
Streptomycin	10 μg	18–41	40 (100)	0	0	80≥	ND	ND
Colistin	10 μg	0	0	0	40 (100%)	ND	ND	ND
Trimethoprim-Sulfamethoxazole	1.25/23.75 μg	15–35	39 (97.5)	1 (2.5%)	0	≥16	11–15	≤10
Gentamicin	10 μg	22–45	40 (100)	0	0	≥16	ND	ND
Ampicillin-sulbactam	20 μg	13–45	26 (65)	1 (2.5%)	13 (32.5%)	≥20	ND	≤19

ND: not determined by CLSI standards. S, Sensitive; I, Intermediate and R, Resistant.

**Table 4 pathogens-12-00082-t004:** Associated virulence and pathogenicity factors found in all Iranian *Brucella* genomes.

Virulence and Pathogenicity Factors	Related Genes
LPS (lipopolysaccharide) pathogenicity factors, entry, intracellular survival and immunomodulatory	*acpXL*, *fabZ*, *gmd*, *htrB*, *kdsA*, *kdsB*, *lpsA*, *lpsB*. *lpcC*, *lpxA*, *lpxB*, *lpxC*, *lpxD*, *lpxE*, *manA*_oAg_, *manC*_oAg_, *per*, *pgm*, *pmm*, *wbdA*, *wbkA*, *wbkB*, *wbkC*, *wboA*, *wbpL*, *wbpZ*, *wzm*, *wzt*.
Type IV secretion system effector secretion	*virB1*, *virB2*, *virB3*, *virB4*, *virB5*, *virB6*, *virB7*, *virB8*, *virB9*, *virB10*, *virB11*, *virB12*.
TIR domain-containing protein immune evasion	*btpA*, *btpB*
Rab2 interacting conserved protein A intracellular survival	*RicA*
CβG (cyclic β-1,2 glucan) intracellular survival	*Cgs*

## Data Availability

All study data are included in the article and Appendix A. The data have also been submitted to the European Nucleotide Archive (ENA). The project accession number is PRJEB50179.

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
