# Peer review of "Determination of Virulence-Associated Genes and Antimicrobial Resistance Profiles in Brucella Isolates Recovered from Humans and Animals in Iran Using NGS Technology"

_pathogens, 2023, doi:10.3390/pathogens12010082_

Round 1
Reviewer 1 Report
Minor language editing may be needed in the introduction and conclusion i.e., line 72 "few/some" - I suggest you pick one; line 80 - "at all" is unnecessary; line 318 - "in spite of" is not correct, rather start the sentence with "The study of resistance....".
You state in line 80 that no data on brucellosis resistance genes from Iran exist but a casual search revealed a number, some of which you site (ref 75, 76). This sentence needs to be more specific to what you mean.
How were the 40 isolates selected? It seems that there would be many more over a 5-year period if the disease is common? It is important that the reader understands how these 40 relate to brucellosis cases (human and animal) in Iran over the time period.
Reference 2 and 4 are the same
Author Response
Dear Editor,
The authors would like to thank the scientific reviewers whose constructive comments have allowed us to improve this research article (Manuscript ID: pathogens-2078098). Please find below our point per point responses to the reviewer’s comments. We have addressed all issues mentioned in the review report and the modifications proposed are highlighted in the revised text.
Respectfully,
Maryam Dadar, Gamal Wareth
One behalf of all authors
Manuscript Number: Manuscript ID: pathogens-2078098
Title: Determination of virulence-associated genes and antimicrobial resistance profiles in Brucella isolates recovered from humans and animals in Iran using NGS technology
Journal: Pathogens
Reviewer 1: Minor language editing may be needed in the introduction and conclusion i.e., line 72 "few/some" - I suggest you pick one; line 80 - "at all" is unnecessary; line 318 - "in spite of" is not correct, rather start the sentence with "The study of resistance....".
Thank you for your precious evaluation. These items changed accordingly.
You state in line 80 that no data on brucellosis resistance genes from Iran exist but a casual search revealed a number, some of which you site (ref 75, 76). This sentence needs to be more specific to what you mean.
The sentence edited as ‘No data on resistance genes of Brucella isolates based on WGS exist from Iran’.
How were the 40 isolates selected? It seems that there would be many more over a 5-year period if the disease is common? It is important that the reader understands how these 40 relate to brucellosis cases (human and animal) in Iran over the time period.
As mentioned in the line of 97-117, the Razi Vaccine and Serum Research Institute, Karaj, Iran as a reference lab of brucellosis in Iran received the animal and human samples for brucellosis evaluation. As you mentioned in this comments, we isolated a lot of isolates each year, however, for this study because of the limitation in budget for WGS, we selected the exanimated isolates according to the various species, various biovars and different geographical locations that Brucella spp. Isolated in Iran.
Reference 2 and 4 are the same.
Thank you for pointing this error out. It corrected accordingly.
I congratulate the authors for carrying out this good work on “Determination of Virulence-Associated Genes and Antimicrobial Resistance Profiles in Brucella Isolates Recovered from Humans and Animals in Iran using NGS Technology”. Brucellosis remains an important zoonosis in many regions of the globe and such comprehensive studies will generate pieces of evidence regarding the epidemiological characteristics of this pathogen. Please find some of the minor comments to be incorporated in the revised manuscript.
Thank you for your precious evaluation and comments.

Reviewer 2 Report
I congratulate the authors for carrying out this good work on “Determination of Virulence-Associated Genes and Antimicrobial Resistance Profiles in Brucella Isolates Recovered from Humans and Animals in Iran using NGS Technology”. Brucellosis remains an important zoonosis in many regions of the globe and such comprehensive studies will generate pieces of evidence regarding the epidemiological characteristics of this pathogen. Please find some of the minor comments to be incorporated in the revised manuscript.
Comments:
Abstract line 02: ‘Imminemet” Please correct the spelling
Line 50: ‘mainly hosts’ can be rewritten as ‘main hosts’
Line 73: the reference is missing for the line “and an increasing trend in the number of Brucella isolates with a resistant phenotype was noted”
Line 75: “or results as consequences from the infected sites porse” – this is not clear to me
Line 76: “Furthermore, the use of virulence factors is still useless to evaluate the pathogenicity of brucellae”- this doesn’t make sense
Line 80: “No data on resistance genes of Brucella isolates exist from Iran at all.”- looks like a blanket statement
Line 108: “All samples were subjected to isolation”: The author mentioned that the study has been carried out on recovered isolates (line 94). Please correct it accordingly.
Line 128: The word “bacteria” can be replaced with ‘isolates’
Line 129: Please mention the criteria for choosing these nine antibiotics
Line 188: “All strains were not susceptible to colistin”- this is repetitive
Table 3: Please keep the second row in the correct format.
Line 233: The use of the word ‘worrying’ can be omitted.
Line 237: I don’t think that brucellosis is a ‘neglected zoonotic disease’, even in developing countries, as lots of awareness are there among farmers for the disease.
Line 296: “may faster the resistance development of these bacteria”: rephrase the sentence
Line 298: “No studies were reported from Iran on samples of human and animal origin.”: This needs to be confirmed by the authors
Author Response
Reviewer 2 Comments:
Abstract line 02: ‘Imminemet” Please correct the spelling
Corrected accordingly.
Line 50: ‘mainly hosts’ can be rewritten as ‘main hosts’
Corrected accordingly.
Line 73: the reference is missing for the line “and an increasing trend in the number of Brucella isolates with a resistant phenotype was noted”
The associated references were added as bellow:
- Marianelli, C.; Graziani, C.; Santangelo, C.; Xibilia, M.T.; Imbriani, A.; Amato, R.; Neri, D.; Cuccia, M.; Rinnone, S.; Di Marco, V. Molecular epidemiological and antibiotic susceptibility characterization of Brucella isolates from humans in Sicily, Italy. J. Clin. Microbiol. 2007, 45, 2923-2928.
- Skalsky, K.; Skalsky, K.; Yahav, D.; Bishara, J.; Pitlik, S; Leibovici, L.; Paul, M. Treatment of human brucellosis: systematic review and meta-analysis of randomised controlled trials. Bmj 2008, 336, 701-704
Line 75: “or results as consequences from the infected sites porse” – this is not clear to me
The sentence rephrased again and edited as below:
However, it is unclear whether this is due to the development of intrinsic or acquired resistance against antibiotic compounds or results as consequences from the intracellular nature of brucellae and thus the inability of antimicrobial compounds to penetrate the infected sites
Line 76: “Furthermore, the use of virulence factors is still useless to evaluate the pathogenicity of brucellae”- this doesn’t make sense
The sentence changed as bellow:
Furthermore, identification and characterization of virulence genes are essential to implementing efficient disease control and prevention approaches and evaluate the pathogenicity of brucellae
Line 80: “No data on resistance genes of Brucella isolates exist from Iran at all.”- looks like a blanket statement
The sentence edited as bellow:
No data on resistance genes of Brucella isolates based on WGS exist from Iran.
Line 108: “All samples were subjected to isolation”: The author mentioned that the study has been carried out on recovered isolates (line 94). Please correct it accordingly.
The procedure of isolation during this study has been reported during the section 2-3 of material and methods to understandable the source of bacteria for reader. However, the sentence edited for more clarification.
Line 128: The word “bacteria” can be replaced with ‘isolates’
Changed accordingly.
Line 129: Please mention the criteria for choosing these nine antibiotics
The criteria have been added as below:
The criteria for choosing these nine antibiotics was based on WHO guidelines on treatment of human brucellosis and the recommendation of used antibiotics in different studies
Line 188: “All strains were not susceptible to colistin”- this is repetitive
Deleted accordingly.
Table 3: Please keep the second row in the correct format.
The table corrected
Line 233: The use of the word ‘worrying’ can be omitted.
Deleted accordingly.
Line 237: I don’t think that brucellosis is a ‘neglected zoonotic disease’, even in developing countries, as lots of awareness are there among farmers for the disease.
The word of neglected deleted from the a ‘neglected zoonotic disease’
Line 296: “may faster the resistance development of these bacteria”: rephrase the sentence
The sentence changed accordingly.
Furthermore, the intracellular lifestyle of brucellae that prohibits the penetration of different antimicrobials into the cells may affect on the resistance development of these bacteria
Line 298: “No studies were reported from Iran on samples of human and animal origin.”: This needs to be confirmed by the authors
Deleted this sentence.
This study is the first WGS analysis for evaluation virulence and resistance genes of Brucella spp.
